# Immune-Stimulating Potential of *Lacticaseibacillus rhamnosus* LM1019 in RAW 264.7 Cells and Immunosuppressed Mice Induced by Cyclophosphamide

**DOI:** 10.3390/microorganisms11092312

**Published:** 2023-09-13

**Authors:** Yeji You, Sung-Hwan Kim, Chul-Hong Kim, In-Hwan Kim, YoungSup Shin, Tae-Rahk Kim, Minn Sohn, Jeseong Park

**Affiliations:** 1Microbiome R&D Center, Lactomason Co., Ltd., Jinju 52840, Republic of Korea; yjyou@lactomason.com (Y.Y.); trkim@lactomason.com (T.-R.K.); ms@lactomason.com (M.S.); 2Food Research Center, Binggrae Co., Ltd., Namyangju 12253, Republic of Korea; ksh1224@bing.co.kr (S.-H.K.); cms51@bing.co.kr (C.-H.K.); tonan@bing.co.kr (I.-H.K.); choksael@bing.co.kr (Y.S.)

**Keywords:** probiotics, *Lacticaseibacillus rhamnosus*, immune enhancement, dietary supplements, immunosenescence

## Abstract

Probiotics, including *Lacticaseibacillus rhamnosus* (*L. rhamnosus*), have gained recognition for their potential health benefits, such as enhancing immune function, maintaining gut health, and improving nutrient absorption. This study investigated the effectiveness of *L. rhamnosus* LM1019 (LM1019) in enhancing immune function. In RAW 264.7 cells, LM1019 demonstrated dose-dependent immune stimulation by increasing nitric oxide production, gene expression of proinflammatory cytokines, and the expression of inducible nitric oxide synthase (iNOS) and cyclooxygenase-2 (COX-2). These effects were mediated through the activation of mitogen-activated protein kinases (MAPKs) and nuclear factor-kappa B (NF-κB) translocation without inducing cytotoxicity. Furthermore, orally administered LM1019 was evaluated in immunosuppressed mice induced by cyclophosphamide (CTX). High-dose administration of LM1019 significantly increased the subpopulations of lymphocytes, specifically helper T cells (CD4^+^), as well as two subtypes of natural killer (NK) cells, namely, IFN-γ^+^ and granzyme B^+^ NK cells. Additionally, LM1019 at a high dose led to elevated levels of proinflammatory cytokines, including IFN-γ and IL-12, compared to CTX-treated mice. These findings highlight the potential of LM1019 in enhancing the immune system. The study contributes to the growing body of research on the beneficial effects of probiotics on immune function.

## 1. Introduction

Immunosenescence, the gradual deterioration of the immune system due to natural aging, heightens susceptibility to infections by various pathogens, particularly affecting older adults [1,2]. Despite various clinical attempts, no approved therapy currently exists for combating immunosenescence [3]. Even though interferons and interleukins have shown potential in treating immune deficiencies, their use in healthy individuals has led to diverse adverse effects and limited success [4,5,6,7]. Therefore, the pressing task is to develop safe and effective food supplements to boost baseline immunity and improve immune function in older adults dealing with immunosenescence.

Probiotics, live microorganisms administered in adequate amounts, have gained significance in addressing various health conditions [8,9]. Probiotics work through multifaceted mechanisms, including maintaining immune homeostasis, influencing gut microbiota, aiding nutrient absorption, and controlling pathogens [10,11,12]. Recent advancements in immunology have highlighted the ability of probiotic species to enhance immune function. Research has demonstrated that probiotics can modulate both innate and adaptive immune responses, leading to the activation of innate immune cells, such as macrophages and dendritic cells [13,14,15]. Moreover, certain strains of probiotics are recognized for their ability to enhance adaptive immunity by stimulating B cells to increase antibody production [16], consequently reducing the risk of infections caused by harmful pathogens [17].

Probiotics possess the capability to activate the immune system through various mechanisms. One such mechanism involves triggering innate pattern recognition receptors, including Toll-like receptors (TLRs), nucleotide-binding oligomerization domain-like receptors, and C-type lectin receptors [13,18,19]. Additionally, several factors within lactobacilli have been identified to influence immune responses both in vitro and in vivo. Adherence factors such as surface carbohydrates, proteins, and glycoproteins involved in colonization of the host epithelium trigger an immune response [20,21,22,23]. Lipoteichoic acids, a major component of the cell wall of gram-positive bacteria, which the majority of probiotic species belong to, act as ligands for TLR-mediated signaling [24,25]. Furthermore, various metabolites secreted from live bacteria have shown specific roles in the host immune system [26,27].

One promising probiotic species in this context is *Lacticaseibacillus rhamnosus* (formerly classified as *Lactobacillus rhamnosus*). While most studies on *L. rhamnosus* strains have reported immunomodulatory roles, primarily alleviating inflammatory conditions, relatively few have delved into their immunostimulatory potential. Han et al. suggested a possible dual role for *L. rhamnosus* strains in immune function to maintain host homeostasis [28]. However, the applicability of these findings to other strains of *L. rhamnosus* remains uncertain.

To further explore the immunostimulatory properties of *L. rhamnosus*, this study focuses on *L. rhamnosus* LM1019 (LM1019). Initially isolated from an infant’s feces in the Republic of Korea as the A-18 strain, LM1019 has demonstrated proteolytic activity against milk casein [29]. Research has revealed that LM1019 possesses excellent probiotic properties and is currently in development as a functional probiotic candidate. In animal studies, LM1019 has exhibited potential in reducing fat cells, neutral fat, subcutaneous fat, and cholesterol, offering promise in preventing and treating obesity. Additionally, it has the capability to lower blood glucose and insulin levels, making it a potential candidate for diabetes prevention and treatment by inhibiting insulin resistance [30]. In this study, the primary objective is to investigate LM1019’s immunostimulatory effects while minimizing immunomodulation. To achieve this, we employed a macrophage cell line and a mouse model to explore LM1019’s immune system-stimulating properties. Our research aims to explore LM1019’s potential to enhance immune responses in RAW 264.7 cells and to investigate the underlying signaling pathways responsible for immune stimulation. Furthermore, we have conducted experiments involving the oral administration of LM1019 to mice with transient immunosuppression. These experiments were designed to assess the impact of LM1019 on splenocyte populations, with a specific focus on natural killer (NK) cells and proinflammatory cytokines.

## 2. Materials and Methods

### 2.1. Cell Line Experiments

#### 2.1.1. Cell Culture

The RAW 264.7 cell line, a murine macrophage cell line, was acquired from the Korean Cell Line Bank (Seoul, Republic of Korea). The cells were cultured in Dulbecco’s Modified Eagle’s Medium (DMEM) (Welgene Inc., Gyeongsan, Gyeongsangbuk-do, Republic of Korea) supplemented with 10% fetal bovine serum (FBS) (Welgene Inc., Gyeongsan, Gyeongsangbuk-do, Republic of Korea) and 1% penicillin–streptomycin solution (Welgene Inc., Gyeongsan, Gyeongsangbuk-do, Republic of Korea). The cells were maintained at 37 °C in a 5% CO_2_ humidified incubator and sub-cultured every 2–3 days.

#### 2.1.2. Cell Viability

To assess the cytotoxicity of LM1019 on RAW 264.7 cells, a 3-(4,5-dimethylthiazol-2-yl)-2,5-diphenyltetrazolium bromide (MTT) assay was conducted following established protocols [31]. RAW 264.7 cells were seeded onto a 96-well plate at a density of 3 × 10^4^ cells/well and incubated for 24 h in a 5% CO_2_ humidified incubator. After removing the cell culture media, cells were exposed to serially diluted doses of LM1019 (1 × 10^7^, 2 × 10^7^, 5 × 10^7^, 1 × 10^8^) or 10 ng/mL Lipopolysaccharide (LPS) from *Escherichia coli* O111:B4 (cat. no. L4391-1MG, Sigma-Aldrich Co., St. Louis, MO, USA). The cells were then incubated for an additional 24 h at 37 °C in a 5% CO_2_ humidified incubator. Following the incubation period, the supernatants were discarded, and the cells were washed twice with 1 × phosphate-buffered saline (PBS). Subsequently, 100 μL of 0.5 mg/mL MTT solution (cat. no. M2128-100 mg, Sigma-Aldrich Co., St. Louis, MO, USA) prepared in complete media was added to each well. The plate was incubated for 2 h at 37 °C. After incubation, the supernatant was carefully removed, and the insoluble formazan crystals formed by viable cells were dissolved by adding 100 μL of dimethyl sulfoxide. The absorbance of the resulting solution was measured at 570 nm using a microplate reader (SpectraMax iD3, Molecular devices, San Jose, CA, USA). Cell viability was calculated using the following Equation (1):(1)$$\mathrm{Cell}\;\mathrm{viability}\;\left(\%\right)=\frac{{\mathrm{OD}}_{570}\;\mathrm{of}\;\mathrm{test}\;\mathrm{group}}{{\mathrm{OD}}_{570}\;\mathrm{of}\;\mathrm{negative}\;\mathrm{control}\;}\times 100$$

#### 2.1.3. The Measurement of Nitric Oxide Generation (Griess Assay)

RAW 264.7 cells were seeded at a density of 3 × 10^4^ cells/well in a 96-well plate and incubated for 24 h in a 5% CO_2_ incubator. After removing the cell culture media, the cells were treated with the indicated doses of LM1019 and/or LPS as follows:Stimulatory Effect Measurement: Serially diluted LM1019 doses (1 × 10^7^, 2 × 10^7^, 5 × 10^7^, 1 × 10^8^ CFU/mL) were applied to the cells to assess their stimulatory effect. To determine the relative intensity, a concentration of 10 ng/mL LPS was used as the positive control for immune stimulation.Inhibitory Effect Evaluation: The cells were pre-incubated with serially diluted LM1019 (10^6^, 10^7^, 10^8^ CFU/mL) for 2 h, followed by the addition of LPS to each well (final concentration = 100 ng/mL) except for the negative control.

After 24 h of incubation at 37 °C in a 5% CO_2_ incubator, the concentration of nitric oxide in the supernatant was determined using the Griess reaction system (cat. no. G2930, Promega Co., Madison, WI, USA) according to the manufacturer’s instructions [32]. The optical density at 540 nm was measured using a microplate reader (SpectraMax iD3, Molecular devices, San Jose, CA, USA).

#### 2.1.4. Quantitative Real-Time Polymerase Chain Reaction (qRT-PCR)

To confirm the change in expression levels of cytokines and immune-related proteins, a two-step RT-PCR was performed. RAW 264.7 cells were seeded on a 6-well plate at 9 × 10^5^ cells/well and incubated for 24 h at 5% CO_2_. After removing the cell culture media, RAW 264.7 cells were treated with serially diluted doses of LM1019 (1 × 10^6^, 2 × 10^6^, 5 × 10^6^, 1 × 10^7^) or 10 ng/mL LPS as a positive control for immune stimulation. Following a 24 h incubation at 37 °C in a 5% CO_2_ incubator, the cell culture media was removed and the cells were washed twice with 1 × PBS. Total RNA was isolated from the cells using the MiniBEST Universal Extraction Kit (cat. no. 9767A, Takara Bio Inc., Shiga, Japan) according to the manufacturer’s instructions. Subsequently, the isolated RNA was then reverse transcribed into cDNA using the High-Capacity cDNA Reverse Transcription Kits (cat. no. 4369914, Thermo Fisher Scientific Inc., Waltham, MA, USA) on a PCR thermal cycler (TaKaRa Bio Inc., Shiga, Japan). The thermocycling conditions were as follows: 25 °C for 10 min and 37 °C for 120 min, followed by denaturation at 85 °C for 5 min. Real-time PCR was then performed using 20 ng of cDNA and SYBR Green PCR Master Mix (cat. no. 4364346, Thermo Fisher Scientific Inc., Waltham, MA, USA) on a QuantStudio 3 real-time PCR system (Applied Biosystems, Waltham, MA, USA). The thermocycling conditions for the real-time PCR were as follows: 95 °C for 10 min and 40 cycles of denaturation at 95 °C for 15 s, followed by annealing and extension at 60 °C for 1 min. The relative expression level of the target gene was calculated using the 2^−ΔΔCt^ method according to Equations (2)–(4), with β-actin serving as the endogenous housekeeping gene [33]. The primers used for this experiment are listed in Table 1.
(2)$$\Delta \mathrm{Ct}={\mathrm{Ct}}_{\mathrm{gene}\;\mathrm{of}\;\mathrm{interest}}-{\;\mathrm{Ct}}_{\mathrm{housekeeping}\;\mathrm{gene}}$$
(3)$$\Delta \Delta \mathrm{Ct}={\Delta \mathrm{Ct}}_{\mathrm{sample}}-{\;\Delta \mathrm{Ct}}_{\mathrm{negative}\;\mathrm{control}}$$
(4)$$\mathrm{Fold}\;\mathrm{difference}=2^{-\Delta \Delta \mathrm{Ct}}$$

#### 2.1.5. Protein Analysis

The protein expression level changes induced by LM1019 in RAW 264.7 cells were assessed using automated capillary western blot assays with the Jess™ system (ProteinSimple, Bio-Techne, Minneapolis, CA, USA) [34,35,36]. RAW 264.7 cells were seeded at a density of 9 × 10^5^ cells/well in a 6-well plate and incubated for 24 h at 37 °C in a 5% CO_2_ incubator. After removing the culture media, the cells were treated with serially diluted doses of LM1019 or 10 ng/mL LPS as the positive control for immune stimulation. Following 24 h of incubation at 37 °C in a 5% CO_2_ incubator, the cell culture media was removed and the cells were washed twice with 1 × PBS. The cells were then lysed using RIPA buffer (cat. no. BRI-9001, Tech&Innovation, Chuncheon, Gangwon-do, Republic of Korea) and diluted to a concentration of 0.5 μg/μL. The cell lysate was mixed with fluorescent 5× master mix (cat. no. SM-W001, ProteinSimple Inc., Santa Clara, CA, USA) at a 4:1 ratio and heated at 95 °C for 5 min to prepare the samples. The samples, along with diluted antibodies were loaded on a 12–230 kDa Jess/Wes Separation Module (cat. no. SM-W001, ProteinSimple Inc., Santa Clara, CA, USA). The primary antibodies used in this analysis are shown in Table 2. HRP-conjugated anti-rabbit secondary antibody was used in the Jess/Wes Separation Module (cat. no. DM-001, ProteinSimple Inc., Santa Clara, CA, USA), and the HRP signal was visualized using peroxide/luminol-S (cat. no. DM-001, ProteinSimple Inc., Santa Clara, CA, USA). The chemiluminescent digital images in the capillary were automatically evaluated by Compass Simple Western version 4.1.0 software (ProteinSimple Inc., Santa Clara, CA, USA).

#### 2.1.6. NF-κB Translocation

The translocation of NF-κB from the cytosol to the nucleus was investigated using an immunofluorescence (IF) assay [37]. RAW 264.7 cells were seeded on a 20Ø confocal dish at a density of 3 × 10^5^ cells/dish and incubated for 24 h at 37 °C in a 5% CO_2_ incubator. After removing the cell culture media, the cells were exposed to serially diluted LM1019 doses (1 × 10^7^, 5 × 10^7^, 1 × 10^8^ CFU/mL) or 10 ng/mL LPS as a positive control for NF-κB nucleus translocation. Following 24 h of incubation at 37 °C in a 5% CO_2_ incubator, the cell culture media was removed, and the cells were double washed with 1 × PBS. The cells were fixed with 4% paraformaldehyde for 10 min at RT, rinsed twice with 1 × PBS, and permeabilized with 0.25% Triton X-100 in PBS for 15 min. Subsequently, the cells were blocked with 1% bovine serum albumin in PBS for 1 h at RT, followed by overnight incubation at 4 °C with a rabbit anti-NF-κB p65 antibody (1:400; cat. no. 8242, Cell Signaling Technology Inc., Danvers, MA, USA). The next day, a FITC-conjugated anti-rabbit antibody (1:2000; cat. no. 656111, Invitrogen, Waltham, MA, USA) was applied for 1 h at RT. The cells were washed twice with 1 × PBS and stained with 200 nM DAPI (cat. no. MBD0015, Sigma-Aldrich Co., St. Louis, MO, USA) for 10 min at RT to label the nucleus, followed by another PBS wash. Finally, cover slides were mounted, and the cell images were captured using an Observer D1 inverted microscope (Carl Zeiss, Oberkochen, Germany) and ZEN 3.1 (blue edition) software (Carl Zeiss, Oberkochen, Germany) with a 100×/1.0 objective lens.

### 2.2. Animal Experiments

#### 2.2.1. Animals

Animal experiments were conducted at KNOTUS, a nonclinical contract research organization in Incheon, Republic of Korea. Six-week-old male BALB/c mice were sourced from Koatech, Gyeonggi-do, Republic of Korea. The mice were maintained under specific conditions: a 12 h light/12 h dark cycle, 23 ± 3 °C temperature, 55 ± 5% humidity, and illuminance of 150 to 300 Lux. All animal care and experimental procedures adhered to approved protocols by the Animal Care and Use Committee (IACUC) of KNOTUS (IACUC no. KNOTUS IACUC 22-KE-0275).

#### 2.2.2. Experimental Design and LM1019 Treatment

After a seven-day stabilization period, mice with an average weight of 22.68 were randomly divided into five groups (five mice per group). The assignment ensured even weight distribution across groups as follows (see Figure 1):Normal control group (NC)Model control group (MC): CTX-induced immunosuppressionCTX + beta-glucan group (PC): CTX-induced immunosuppression + beta-glucan 80 mg/kgCTX + LM1019 low-dose group (LM-L): CTX-induced immunosuppression + 10^8^ CFU/headCTX + LM1019 high-dose group (LM-H): CTX-induced immunosuppression + 10^9^ CFU/head

Immunosuppression was induced by intraperitoneal injection of 150 mg/kg CTX on Day 0 and Day 2, except for the normal control group. From Day 3 to Day 7, the test materials (beta-glucan and LM1019) were orally administered once daily for five days. Mouse body weights were recorded on the day of immunosuppression initiation and daily throughout the experiment. On Day 8, mice were sacrificed following overnight fasting and anesthesia using isoflurane. After confirming anesthesia, mice were euthanized, and blood and spleen samples were collected for further analysis, including fluorescence-activated cell sorting (FACS) analysis.

#### 2.2.3. Change in Lymphocyte Activity

The activity of NK cells, CD4^+^ T cells, and CD8^+^ T cells, as well as the production of IFN-γ and granzyme B by NK cells, were analyzed using FACS analysis with the harvested spleen tissues [37,38,39]. Spleen tissue was obtained, and splenocytes were prepared using a mouse spleen dissociation kit (cat. no. 130-095-926, Miltenyi Biotec, Bergisch Gladbach, Germany) following the manufacturer’s instructions. Subsequently, red blood cell lysis was performed, and the cells were resuspended in RPMI 1640 media. The splenocytes were then seeded onto round-bottom 96-well plates at a density of 1 × 10^6^ cells/well.

For NK cell activation, a mixture of cell stimulation cocktail (cat. no. 00-4970-93, Invitrogen, Waltham, MA, USA), monensin solution (cat. no. 00-4505-51, Invitrogen, Waltham, MA, USA), and brefeldin A solution (cat. no. 420601, BioLegend, San Diego, CA, USA) was added to the cells, followed by incubation for 3 h at 37 °C in a 5% CO_2_ incubator. To analyze the T cell population, the splenocytes were treated with an antibody mix containing FITC-CD3 (cat. no. 11-0032-82, Invitrogen, Waltham, MA, USA), PE-CD4 (cat. no. 12-0041-82, Invitrogen, Waltham, MA, USA), APC-CD8a (cat. no. 17-0081-82, Invitrogen, Waltham, MA, USA), and APC-eFluor™ 780-CD45 (cat. no. 47-0451-82, Invitrogen, Waltham, MA, USA) antibodies in flow cytometry staining buffer (cat. no. 00-4222-57, Invitrogen, Waltham, MA, USA). After incubation in a dark place at 4 °C for 30 min, the splenocytes were washed twice with 1 × PBS and FACS analysis was performed using Novocyte 3000 (Agilent Technologies Inc., Santa Clara, CA, USA).

For NK cell activity analysis, both surface staining and intracellular staining were performed. Surface staining was conducted using a buffer and antibody mix containing flow cytometry staining buffer, monensin solution, brefeldin A solution, Super Bright™ 600-CD3 (cat. no. 63-0031-82, Invitrogen, Waltham, MA, USA), PE-Cyanine7-CD335 (cat. no. 25-3351-82, Invitrogen, Waltham, MA, USA), FITC-CD49b (cat. no. 11-5971-82, Invitrogen, Waltham, MA, USA), and APC-eFluor™ 780-CD45 (cat. no. 47-0451-82, Invitrogen, Waltham, MA, USA) antibodies. After staining in a light-blocked environment at 4 °C for 30 min, the splenocytes were washed twice with permeabilization buffer (cat. no. 00-8333-56, Invitrogen, Waltham, MA, USA). For intracellular staining, permeabilization buffer was used in combination with APC-IFN-γ (cat. no. 17-7311-82, Invitrogen, Waltham, MA, USA) and PE-granzyme B (cat. no. 12-8898-82, Invitrogen, Waltham, MA, USA) antibodies. The cells were subjected to intracellular staining at RT for a duration of 20 min. Finally, FACS analysis was conducted to analyze the stained cells.

#### 2.2.4. Serum Cytokine Level Change

The concentrations of IFN-γ, IL-12, and IL-2 in the serum were analyzed using frozen serum samples and a commercially available sandwich ELISA kit. IFN-γ was measured using the MIF00 kit (R&D Systems, Minneapolis, MN, USA). IL-12 was measured using the M1270 kit (R&D Systems, Minneapolis, MN, USA). IL-2 was measured using the M2000 kit (R&D Systems, Minneapolis, MN, USA). The analysis was performed according to the manufacturer’s instructions.

### 2.3. Statistical Analysis

All results were expressed as the mean ± SD. To compare the differences of groups, a one-way analysis of variance (ANOVA) followed by Dunnett’s multiple comparison test was applied. The statistical analysis was performed using Prism 7.04 software (GraphPad Software Inc., Boston, MA, USA), and a *p*-value less than 0.05 was considered statistically significant. The levels of significance were indicated as * *p* < 0.05, ** *p* < 0.01, and *** *p* < 0.001.

## 3. Results

### 3.1. Immune System-Stimulating Effects of LM1019 on RAW 264.7 Cells

#### 3.1.1. LM1019-Mediated Upregulation of Nitric Oxide Production and Gene Expressions of Proinflammatory Cytokines

To determine the effect of LM1019 on immunity in RAW 264.7 cells, we first assessed its cytotoxicity using an MTT assay, a conventional method of measuring intracellular mitochondrial enzymatic activity. LM1019 treatment at doses ranging from 1 × 10^7^ to 1 × 10^8^ CFU/mL did not result in any observed cytotoxicity, as confirmed by the absence of a decrease in the intracellular mitochondrial activity. As a positive control for immune stimulation, we treated the cells with LPS. We then analyzed the production of nitric oxide in cell culture media, which is considered a biomarker of immune response by macrophages [40]. We applied serially diluted LM1019 to RAW 264.7 cells and found that the level of nitric oxide significantly increased in a dose-dependent manner, almost reaching the same level as that observed in LPS-treated cells (10 ng/mL) at high dose of LM1019 (Figure 2b). These findings suggest that LM1019 activates RAW 264.7 cells, leading to an immune response. Moreover, as no cytotoxicity was observed at the same doses, we assume that the elevation of nitric oxide was not induced by cellular stress and damage (Figure 2a).

Next, we assessed the immunomodulatory effect of LM1019 on RAW 264.7 cells. To address this question, we pretreated the cells with different doses of LM1019 (10^6^, 10^7^, 10^8^ CFU/mL), followed by the application of LPS to stimulate nitric oxide production. We expected to observe an inhibitory effect on LPS-induced nitric oxide generation which has been reported in most *L. rhamnosus* strains. However, surprisingly, we only observed minor inhibitory effects at low and medium doses of the LM1019 treatment, and there was no dose-dependency in these two doses. Furthermore, no inhibition of LPS-induced nitric oxide generation was observed at high dose of LM1019 (Figure 2c). These findings suggest that LM1019 does not exert a potent immunomodulatory effect on RAW 264.7 cells.

To confirm that the elevated levels of nitric oxide observed corresponded to an increase in proinflammatory cytokines caused by LM1019, we examined the expression levels of IL-1β, IL-6, and TNF-α through qRT-PCR. After treatment with LM1019, RNAs were isolated and reverse transcribed to cDNA. As shown in Figure 2d–f, treatment with LM1019 resulted in dose-dependent exponential increases in mRNA expression levels of IL-1β, IL-6, and TNF-α. Compared to the negative control, IL-1β expression increased up to 688.7 times (Figure 2d), IL-6 up to 13.7 times (Figure 2e), and TNF-α up to 20.9 times (Figure 2f). These findings clearly indicate that treatment of LM1019 led to a strong immune response in RAW 264.7 cells.

#### 3.1.2. Overexpression of Proinflammatory Proteins and Signaling Pathway for Immune Stimulation by LM1019

LM1019 was found to have an immune system-stimulating effect by promoting the production of nitric oxide and proinflammatory cytokines. To assess the effects of LM1019, changes in mRNA and protein levels of iNOS and COX-2 were monitored, as iNOS synthesizes nitric oxide for immune responses [41] while COX-2 is expressed in response to inflammatory stimuli such as increased cytokines, growth factors, and TGF-β [42]. LM1019 treatment (1 × 10^6^, 2 × 10^6^, 5 × 10^6^, 1 × 10^7^ CFU/mL) resulted in boosted gene expressions for iNOS and COX-2 in RAW 264.7 cells, with significant differences compared to the control group (no treatment) (for iNOS at 2 × 10^6^, 5 × 10^6^, 1 × 10^7^ CFU/mL; for COX-2 at 5 × 10^6^, 1 × 10^7^ CFU/mL), as shown in Figure 2a, b. Specifically, iNOS mRNA increased by up to 7.5 times (Figure 3a) and COX-2 mRNA increased by up to 75.6 times (Figure 3b) compared to the control at high doses of LM1019. These up-regulations in gene expression were also reflected in the protein levels of iNOS and COX-2 (Figure 3c), suggesting that LM1019 directly activates RAW 264.7 cells, resulting in boosted gene expressions for proinflammatory cytokines, as previously shown in Figure 2.

Next, we aimed to investigate the mechanisms by which LM1019 triggered proinflammatory signaling in these cells by analyzing immune system-related proteins. Based on the known relationship between MAPKs and the production of proinflammatory cytokines [43,44,45,46], we hypothesized that LM1019 potentiates the NF-κB pathway by activating MAPKs. To test this hypothesis, we measured the phosphorylation of extracellular signal-regulated kinase 1/2 (Erk1/2) and Jun N-terminal kinase (JNK), two MAPKs, in RAW 264.7 cells stimulated with varying concentrations of LM1019 for 24 h. As expected, both Erk1/2 and JNK were phosphorylated in response to LPS exposure, and LM1019 also prompted their phosphorylation in a dose-dependent manner (Figure 3d). To further evaluate the role of LM1019 in modulating the downstream targets of Erk1/2 and JNK, we assessed the nuclear translocation of NF-κB. After treating cells with LM1019 for 24 h, we stained both the cytosolic and nuclear forms of NF-κB with anti-NF-κB antibody and counterstained the nucleus with DAPI. Our results showed that LM1019 treatment caused translocation of NF-κB from the cytosol to the nucleus, similar to the activation of MAPKs (Figure 3e).

### 3.2. Immune System-Stimulating Effect of LM1019 in an Immunosuppressive Mouse Model

#### 3.2.1. The Changes in the Weights of the Body and the Spleen

The aim of this study was to investigate the immune system-stimulating effect of LM1019 in an immunosuppressive mouse model. To assess the immune system-enhancing activity, transient immune depletion was induced in BALB/c mice by administering cyclophosphamide (CTX) on Day 0 and Day 2, as described in the methods section. Oral administration of LM1019 started on Day 3 and continued daily until Day 7. For comparison, beta-glucan was used as a positive control for immune system-stimulating activity [47,48,49,50].

Following CTX-induced immune depletion, the mice exhibited overall weight loss, with the total body weights being approximately 10% lower than those of the normal control group from Day 2 to Day 8 after CTX administration (Figure 4a,b). This weight loss is commonly associated with malnutrition due to loss of appetite and inhibition of growth and development in mice [51,52]. Similarly, the spleen weight of the immunosuppression-induced group was significantly lower than that of the normal control group. However, no significant difference was observed between the model control group and the LM1019 groups.

#### 3.2.2. The Changes in Lymphocyte Populations

To evaluate the changes in the spleen caused by LM1019, we monitored the lymphocyte populations in splenocytes after sacrifice. The results indicated that CTX treatment (model control group) significantly downregulated immune functions in the mice, as evidenced by diminished numbers of NK cells, CD4^+^ T cells, and CD8^+^ T cells in the spleen (Figure 5a–c). Additionally, the population of NK cells expressing IFN-γ was severely reduced compared to the control group (Figure 5d). After CTX-induced immune suppression, oral supplementation of beta-glucan resulted in the recovery of immune functions. The numbers of NK cells, CD4^+^ T cells, and CD8^+^ T cells were significantly higher compared to the model control group (Figure 5a–c), indicating the verified stimulatory effects of beta-glucan on the immune system.

Furthermore, the study investigated the immune system-stimulating effects of LM1019. Oral administration of LM1019 after CTX treatment alleviated the damages caused by immune suppression, resulting in a more than 2.2-fold increase in the number of CD4^+^ T cells compared to the model control group, particularly in CTX + high-dose LM1019 groups (Figure 5b). Although statistical differences were not observed, there was a tendency for increased numbers of NK cells and CD8^+^ T cells in LM1019-treated groups, especially at high doses (Figure 5a,c). Moreover, the number of NK cells expressing IFN-γ was significantly increased compared to the model control group (Figure 5d). The role of LM1019 in inducing expression of granzyme B, a serine protease found in activated NK cells, was also assessed. Both the CTX + beta-glucan group and the CTX + high-dose LM1019 group exhibited significantly higher numbers of granzyme B^+^ NK cells compared to the no treatment and model control groups (Figure 5e), indicating that LM1019 boosted the activity of NK cells by increasing granzyme B levels.

#### 3.2.3. Proinflammatory Cytokines in the Serums

The aim of the study was to determine if the administration of LM1019 promotes systemic immune stimulation in the CTX-induced immunosuppression model. To assess this, levels of proinflammatory cytokines in the serum were measured using ELISA. The results showed that CTX treatment (model control group) significantly decreased levels of IFN-γ (40.4%) and IL-12 (34.2%) compared to the normal control group, as expected.

In the presence of CTX, both the CTX + beta-glucan group and CTX + high-dose LM1019 group exhibited a significant increase in IFN-γ levels, with a 2.6-fold and 2.8-fold increase, respectively, compared to the model control group (Figure 6a). Similarly, IL-12 levels, which were reduced by CTX, were restored to almost the levels of the normal control group, with a 3.2 to 3.3-fold increase upon administration of beta-glucan and LM1019 (Figure 6b). IL-2 levels followed a similar trend as IL-12 and IFN-γ, although no statistically significant difference was observed between the groups (Figure 6c).

## 4. Discussion

The aim of this study was to investigate the immune system-stimulating properties of the LM1019 probiotic strain and its potential as a functional health supplement. We conducted experiments using both RAW 264.7 cell lines and an artificial immunosuppressed animal model to evaluate the different effects of LM1019 on the immune system.

In the cell line experiment, we observed several significant outcomes. Firstly, the application of LM1019 resulted in a dose-dependent increase in nitric oxide production without causing any cytotoxicity (Figure 2a,b). Furthermore, LM1019 treatment led to elevated transcriptional levels of proinflammatory cytokines (Figure 2d–f), implying its capacity to enhance the production of critical inflammatory molecules involved in immune defense.

However, it is essential to note that our study did not reveal a significant immunomodulatory effect of LM1019 on this cell line, which contradicts findings for other strains within the same species [53,54,55,56,57,58,59,60,61,62]. Previous research has highlighted the strain-specific effects of probiotics on immune function. For instance, Dong et al. compared six probiotic strains from different species and found differing effects on various immune parameters [63]. López et al. also emphasized the variability in immune cell maturation induced by different strains from the same probiotic species [64]. These studies emphasize that probiotic strains can have distinct impacts on immune function, depending on the specific strain and specific immune parameters assessed. In light of this, we propose that while certain *L. rhamnosus* strains with immunomodulatory properties may be suitable for individuals with inflammatory diseases, LM1019’s unique ability to exhibit an immunostimulatory effect without immunomodulation makes it a promising candidate for individuals with weakened basal immunity.

Our study further confirmed the involvement of the ERK and JNK signaling pathways in LM1019-mediated immune stimulation, and this activation was found to be dose-dependent (Figure 3d). These pathways, which belong to the MAPK family, play pivotal roles in regulating immune responses and are activated by LM1019. As a strain of *L. rhamnosus*, LM1019 is believed to activate TLRs in RAW 264.7 cells. Upon activation, TLRs recruit adaptor proteins, leading to the subsequent activation of downstream kinases and the phosphorylation of transforming growth factor-β-activated kinase 1 (TAK1), an upstream kinase for NF-κB and MAPKs. This cascade ultimately results in the generation of proinflammatory cytokines. Additionally, TLR signaling pathways activate NF-κB-mediated generation of proinflammatory cytokines. TLR-mediated activation of TAK1 results in phosphorylation of IκBα, leading to its ubiquitination-dependent degradation. As a result, IκBα-bound NF-κB (inactive form) is transformed into active NF-κB, which translocates into the nucleus to become a crucial transcription factor for generating proinflammatory cytokines [65,66,67,68]. The findings suggest that LM1019-mediated MAPK activation may be partially attributed to the lipoteichoic acid (LTA) derived from the cell wall components of LM1019. This hypothesis is supported by the observation that various LTAs obtained from different lactic acid bacteria, including *L. rhamnosus*, exhibit immune system-stimulating effects [69]. These effects manifest as an increase in nitric oxide production, upregulation of iNOS expression, and secretion of proinflammatory cytokines in the same cell line. Furthermore, studies conducted using cell-free supernatants from various lactic acid bacteria, including strains of *L. rhamnosus*, which would contain metabolites secreted by bacteria, indicated upregulation of these immune markers in murine and human macrophage-like cell lines [2,3,8,13,14,15,26,70]. From those findings, we infer that the metabolites produced by living LM1019 may potentiate TLR-mediated signaling pathways, subsequently leading to the production of proinflammatory cytokines. These findings highlight the potential role of LM1019-derived LTA and its secreted metabolites in the activation of the MAPK and NF-κB-mediated pathways and the subsequent modulation of immune responses. However, the precise mechanisms underlying these processes require further investigation to elucidate the molecular interactions between LTAs, metabolites, and immune cells.

An animal study was carried out to observe the effects of orally administering LM1019 on the body and immune cells. This was achieved by creating a CTX-induced immunosuppressed mouse model, which mimics weakened immune systems, often observed in the elderly, by using CTX to disrupt DNA synthesis and functionality in T cells and B cells, leading to overall immune function downregulation. Interestingly, both LM1019 and the positive control (beta-glucan) failed to prevent weight loss. However, it is crucial to note that LM1019 administration did not exacerbate the condition, and no adverse effects were observed in the mice. Furthermore, oral administration of LM1019 in immunosuppressed mice enhanced immune function in the spleen (Figure 5). The spleen, being the largest lymphatic organ, plays a vital role in filtering blood, eliminating antigens, and initiating adaptive immune responses [71,72]. Specifically, LM1019 treatment increased the number of T lymphocytes, including NK cells, in the spleen, indicating its potential to enhance immune cell populations. Additionally, LM1019 led to an increased population of NK cells expressing IFN-γ and granzyme B, proinflammatory cytokines, and serine proteases involved in immune defense. This suggests that LM1019 can enhance NK cell activity, further supporting its immune system-stimulating properties. Interestingly, the number of granzyme B^+^ NK cells increased in the model control (CTX-induced immunosuppression only) compared to the normal control. This was unexpected, as two treatments of CTX would typically suppress NK cell maturation. One possible explanation is that CTX-mediated damage caused elevated granzyme B expression in NK cells, which might be needed to kill damaged cells caused by CTX (helping apoptosis). This inference is supported by other reports indicating granzyme B-mediated apoptosis of various cells in stressed conditions [73,74,75]. Despite the elevation of granzyme B^+^ NK cells in the model control, numbers of granzyme B^+^ NK cells were still significantly increased in both the positive control (beta-glucan) and high-dose LM1019 models compared to the model control. This suggests that LM1019 can enhance the activity of NK cells, further supporting its immune system-stimulating properties. Finally, evidence was obtained showing that LM1019 exhibited an immune system-stimulating effect in the harvested serum, as evidenced by elevated levels of proinflammatory cytokines (Figure 6). These findings demonstrate that LM1019 stimulates the secretion of IL-12, a cytokine known for its role in differentiating Th0 cells into Th1 cells, along with IFN-γ and IL-2, predominantly secreted by Th1 cells [76,77]. Furthermore, changes in cytokine secretion were concentration-dependent, suggesting that LM1019 activates the immune system by inducing Th1 cell activation and enhancing the secretion of Th1 cytokines. This indicates that LM1019 has the potential to promote the production of proinflammatory cytokines in systemic circulation, contributing to overall immune system enhancement.

## 5. Conclusions

In conclusion, the findings of this study strongly suggest that LM1019 possesses potential immune system-stimulating properties. The oral administration of LM1019 induced elevated expression of proinflammatory cytokines and promoted the activation of NK cells in an in vivo experiment. Furthermore, in RAW 264.7 cells, we provided evidence of activation of the ERK and JNK signaling pathways by the LM1019 treatment. By visualizing facilitated translocation of NF-κB after LM1019 treatment, MAPK-dependent NF-κB activation for generating proinflammatory cytokines was demonstrated (Figure 4e). While further research, including human trials, is necessary to fully comprehend the underlying mechanisms and potential clinical implications of these findings, results of this study suggest that LM1019 has the potential to be used as a health supplement ingredient for enhancing basal immunity in older adults with immunosenescence.

## Figures and Tables

**Figure 1 microorganisms-11-02312-f001:**
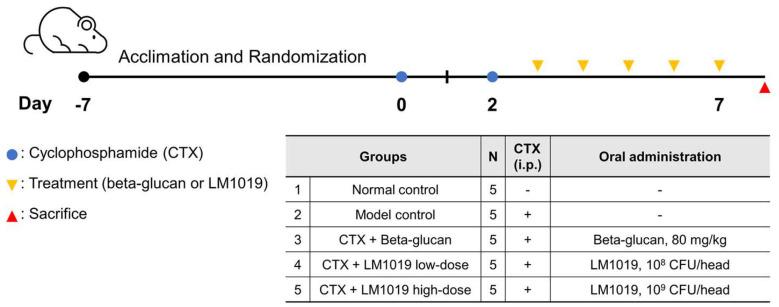
Schematic diagram illustrating the experimental design used to assess the immune system-stimulating activity of LM1019. Immunocompromised BALB/c mice were rendered immunosuppressed by intraperitoneal administration of 150 mg/kg of cyclophosphamide (CTX) twice on Day 0 and Day 2 for the model control, positive control (beta-glucan), and two doses of LM1019 groups. The normal control group received no treatment except regular feeding. After inducing artificial immune suppression in the CTX-treated groups, the CTX + beta-glucan group was orally treated with beta-glucan (80 mg/kg), while the CTX + LM1019 groups were fed two doses of LM1019 (10^8^ or 10^9^ CFU/head) from Day 3 to Day 7. On Day 8, all the mice were sacrificed and biomarkers were analyzed.

**Figure 2 microorganisms-11-02312-f002:**
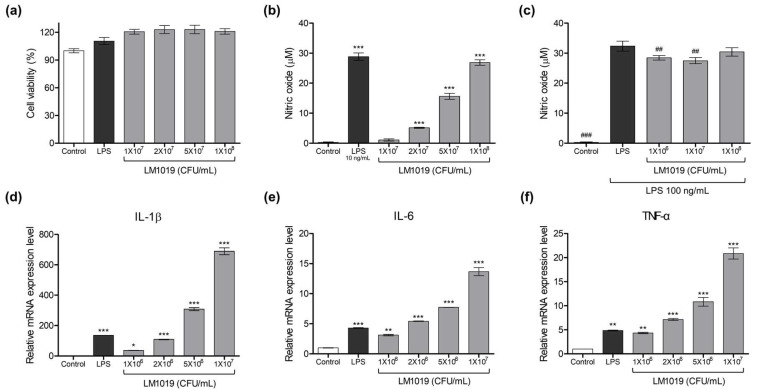
Immune system-stimulating effects of LM1019 on RAW 264.7 cells. (**a**) Cell viability was assessed using an MTT assay after 24 h of treatment with varying doses of LM1019 (1 × 10^7^, 2 × 10^7^, 5 × 10^7^, 1 × 10^8^ CFU/mL) (*n* = 3). (**b**) The stimulatory effect on nitric oxide (NO) generation of LM1019 was evaluated by performing a Griess assay (*n* = 3). (**c**) The inhibitory role of LM1019 on LPS-induced nitric oxide generation was also assessed using a Griess assay (*n* = 3). mRNA expression levels for (**d**) IL-1β, (**e**) IL-6, and (**f**) TNF-α, respectively, were quantified using qRT-PCR and normalized to β-actin expression. RAW 264.7 cells were treated with varying doses of LM1019 (1 × 10^6^, 2 × 10^6^, 5 × 10^6^, 1 × 10^7^ CFU/mL) to evaluate their immune system-stimulating effects (*n* = 2). As a positive control for immune stimulation, LPS (10 ng/mL) was used. The results are presented as mean ± SD. Statistical significance was determined by * *p* < 0.05, ** *p* < 0.01, and *** *p* < 0.001 compared to the control group. Statistical significance was determined by ## *p* < 0.01 and ### *p* < 0.001 compared to the LPS group.

**Figure 3 microorganisms-11-02312-f003:**
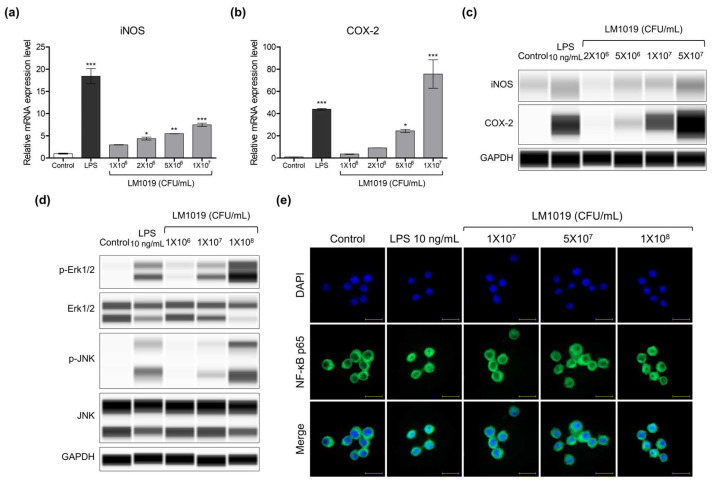
LM1019-induced upregulation of proinflammatory gene expression and proteins and the signaling pathway for NF-κB activation in RAW 264.7 cells. mRNA expression levels of (**a**) iNOS and (**b**) COX-2 were quantified using qRT-PCR and normalized to β-actin expression (*n* = 2). (**c**) Protein levels of iNOS and COX-2 were evaluated by performing western blots using Jess™. (**d**) Activation of the MAPKs pathway by LM1019 was assessed by western blots using Jess™. Phosphorylated Erk1/2 and JNK were visualized to evaluate MAPK activation while the total Erk1/2 and JNK served as comparative controls. GAPDH was used as the loading control. (**e**) LM1019-induced translocation of NF-κB from the cytosol to the nucleus was monitored by immunofluorescent microscopy (Scale bar = 20 μm). Statistical significance was determined by * *p* < 0.05, ** *p* < 0.01, and *** *p* < 0.001 compared to the control group.

**Figure 4 microorganisms-11-02312-f004:**
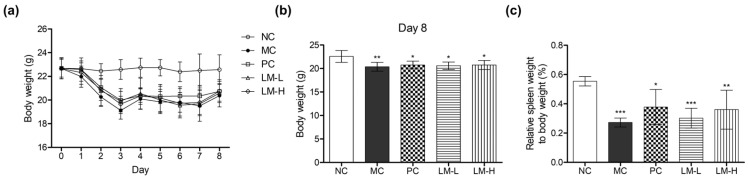
Changes in body and spleen weights during experimental periods. (**a**) Overall body weight change, (**b**) body weight at Day 8, and (**c**) relative spleen weight to body weight. The experimental groups include: NC: normal control group (no treatment), MC: model control group (CTX-induced immunosuppression only), PC: CTX + Beta-glucan group (CTX-induced immunosuppression followed by beta-glucan administration), LM-L: CTX + low-dose LM1019 group (CTX-induced immunosuppression followed by 10^8^ CFU/head of LM1019 administration), LM-H: CTX + high-dose of LM1019 group (CTX-induced immunosuppression followed by 10^9^ CFU/head of LM1019 administration). Data are presented as mean ± SD (*n* = 5). Statistical significance was determined by * *p* < 0.05, ** *p* < 0.01, and *** *p* < 0.001 compared to the NC.

**Figure 5 microorganisms-11-02312-f005:**
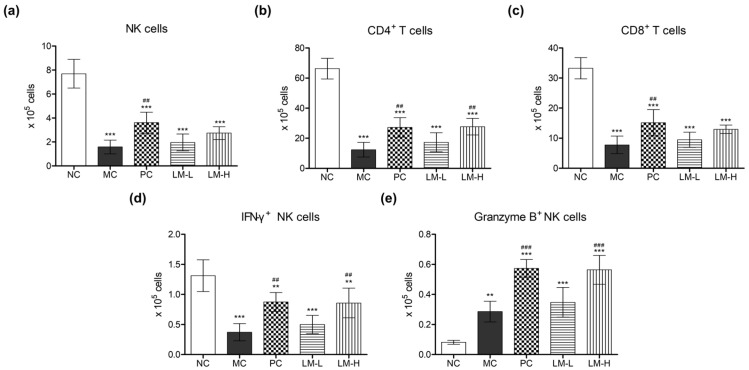
Quantification of NK and T cell populations and assessment of NK cell activities in the spleen. (**a**) Number of NK cells isolated from the spleens. (**b**) Number of CD4^+^ T cells isolated from the spleens. (**c**) Number of CD8^+^ T cells isolated from the spleens. (**d**) Number of IFN-γ^+^ NK cells isolated from the spleens. (**e**) Number of granzyme B^+^ NK cells isolated from the spleens. Lymphocytes were isolated from the harvested spleens after sacrifice, and the numbers of isolated lymphocytes were measured using FACS analysis. The experimental groups include NC: normal control group (no treatment), MC: model control group (CTX-induced immunosuppression only), PC: CTX + beta-glucan group (CTX-induced immunosuppression followed by beta-glucan administration), LM-L: CTX + low-dose LM1019 group (CTX-induced immunosuppression followed by 10^8^ CFU/head of LM1019 administration), LM-H: CTX + high-dose of LM1019 group (CTX-induced immunosuppression followed by 10^9^ CFU/head of LM1019 administration). Data are presented as mean ± SD (*n* = 5). Statistical analysis was performed, and significance levels were indicated as follows: ** *p* < 0.01, and *** *p* < 0.001 against NC; ## *p* < 0.01 and ### *p* < 0.001 against MC.

**Figure 6 microorganisms-11-02312-f006:**
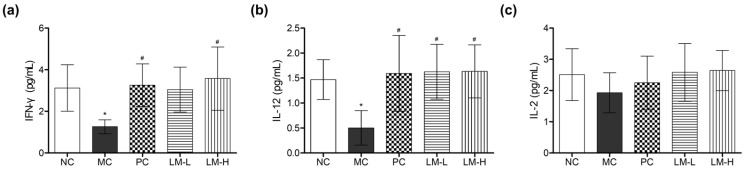
Recovery of proinflammatory cytokines in the serum by LM1019 following CTX-induced immunosuppression. (**a**) Amount of IFN-γ. (**b**) Amount of IL-12. (**c**) Amount of IL-2. The levels of IFN-γ, IL-12, and IL-2 were quantified using ELISA assays on harvested serums from sacrificed mice after the animal experiment. The experimental groups include: NC: normal control group (no treatment), MC: model control group (CTX-induced immunosuppression only), PC: CTX + beta-glucan group (CTX-induced immunosuppression followed by beta-glucan administration), LM-L: CTX + low-dose LM1019 group (CTX-induced immunosuppression followed by 10^8^ CFU/head of LM1019 administration), LM-H: CTX + high-dose of LM1019 group (CTX-induced immunosuppression followed by 10^9^ CFU/head of LM1019 administration). Data are presented as mean ± SD (*n* = 5). Statistical significance levels are indicated as follows: * *p* < 0.05 against NC, and # *p* < 0.05 against MC.

**Table 1 microorganisms-11-02312-t001:** The primer sequence for qRT-PCR.

Gene Name	Accession No.	Nucleotide Sequence	Product Size (bp)
*IL-1β*	NM_008361.4	F: GGG CCT CAA AGG AAA GAA TC R: TAC CAG TTG GGG AAC TCT GC	183
*IL-6*	NM_031168.2	F: AGT TGC CTT CTT GGG ACT GA R: CAG AAT TGC CAT TGC ACA AC	191
*TNF-α*	D84199.2	F: ATG AGC ACA GAA AGC ATG ATC R: TAC AGG CTT GTC ACT CGA ATT	276
*iNOS*	BC062378.1	F: TTC CAG AAT CCC TGG ACA AG R: TGG TCA AAC TCT TGG GGT TC	180
*COX-2*	NM_011198.4	F: AGA AGG AAA TGG CTG CAG AA R: GCT CGG CTT CCA GTA TTG AG	194
*β-Actin*	NM_007393.5	F: CCA CAG CTG AGA GGG AAA TC R: AAG GAA GGC TGG AAA AGA GC	193

**Table 2 microorganisms-11-02312-t002:** The primary antibody information for protein analysis.

Name of Antibody	Company	cat. no.	Source	Dilution
COX-2	Cell Signaling Technology Inc. (Danvers, MA, USA)	2282	Rabbit	1:10
iNOS		13120	Rabbit	1:10
p-Erk1/2		9101	Rabbit	1:10
Erk1/2		9102	Rabbit	1:10
p-JNK		9251	Rabbit	1:10
JNK		9252	Rabbit	1:10
GAPDH		2118	Rabbit	1:50

## Data Availability

Not applicable.

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
