# Peer review of "Immune-Stimulating Potential of Lacticaseibacillus rhamnosus LM1019 in RAW 264.7 Cells and Immunosuppressed Mice Induced by Cyclophosphamide"

_microorganisms, 2023, doi:10.3390/microorganisms11092312_

Round 1

Reviewer 1 Report

Comments to manuscript ID microorganisms-2580440

1.      In the sections of introduction, all background especially the knowledge of L. rhamnosus LM1019 should be added base on your current manuscript. There is more information from published papers that not fully presented or they may not have grasped the author's true meaning, for example: probiotics exert a modulatory effect on dendritic cells, How or how to generate an impact through the content of the author's research; the difference between the role of probiotics and the role of L. rhamnosus LM1019, can the role of probiotics be used to replace the role of L. rhamnosus LM1019, etc. Thus, the manuscript must be re-edited.

2.      In 2.1.2, cells were treated with serially diluted doses of LM1019 (1, 2, 5, 10×107) that is the conventional way of scientific counting, but in your results or figures showing 10×107 becomes 1×108.

3.      In table 1, the NCBI gene ID number and the size of primers should be added. In the first line of table, there is a “gene” name, not “protein”, and should be italic.

4.      In the section of 2.3 in manuscript, the “p” should be changed to the “P” in line 288, including the other paragraphs.

5.      Figure 7 should be moved to the section of methods instead of serving as a summary picture.

6.      The authors must discuss the roles of each protein in MAPK and the interactions between other proteins, including the inflammation. Also explain why we didn't do p38

Extensive editing of English language required for further edit more explain.

Reviewer 2 Report

Exemplary paper, a worthy addition to a fast-growing body of important research. The authors conducted a comprehensive investigation of the dose-dependent immunomodulatory effects of L. rhamnosus LM1019, including cell line and animal studies, mRNA and protein levels, NO production, cytokine levels, lymphocyte populations, and even something about the signaling pathways involved. The methods and the results are described and presented, respectively, with commendable precision, including appropriate controls and statistical analysis. The figures are intuitive almost to the point of making their legends superfluous. Introduction and Discussion are concise and to the point. The supplementary file is also appreciated.

I would suggest only a few minor revisions:

- Line 98: please explain what “LPS” means when first mentioned.

- Lines 162-170: the antibodies, complete with catalogue numbers and dilutions, would look much nicer in table form.

- Lines 307-309: the strain LM1019, identified as the species L. rhamnosus, not “the species LM1019, identified as L. rhamnosus”; please also clarify that it is the species, not the particular strain, that has been shown to have immunomodulatory properties in other studies.

- Lines 422-431: more suitable to the discussion.

- Lines 540-544: it might be worth noting that according to ref. 58 LTA from L. rhamnosus is among the least potent activators of the MAPK pathway, suggesting a different, LTA-independent mechanism or, perhaps, a dose-dependent response to be studied further.

- Lines 563-565: would you provide some explanation about the difference between NK cells with granzyme B and IFN-g (Figure 5d-e)? How come the former are increased in the model control group compared to the normal control group, while the later are decreased, as indeed are the CD4+, CD8+ and NK cells in general? Why should immunosuppression increase the number of NK cells with granzyme B?

The English is acceptable. Only a few cases of mild confusion and some typos need to be addressed. For example:

- Line 332: reverse transcribed to cDNA, not “synthesized to”.

- Lines 345-346: a little clarification is needed here about COX-2; for example, “the expression of COX-2 is stimulated by inflammation, cytokines, growth factors and TGF-b”.

- Line 521: transcriptional levels are increased or decreased, rather than promoted or demoted.

- Lines 563-564: this needs clarification – IFN-g is a proinflammatory cytokine and granzyme B belong is a serine protease involved in the immune response – neither is both at the same time.

- Typos and formatting slips, e.g. missing full stop after the temperature at line 104; “of” that should be “for” in line 294; “ex-pression” in line 334; “increases” in line 323; “varying of doses of” in line 335.

Round 2

Reviewer 1 Report

1. In the section of introduction, the information of Lacticaseibacillus rhamnosus is more, and a sentence or two sentences to show their benefits  is sufficient. So, author should add supplement the immune background or other content mentioned in this article based on your manuscript.

2. In the section of results, the author only needs to describe the changes in the results, without excessive discussion or introduction of knowledge background. Please simplify.

3. The discussion is a further analysis of the results, not a repetitive description of the results. Please correct.

Minor editing of English language

Author Response

Response to Reviewer 1 Comments

  1. In the section of introduction, the information of Lacticaseibacillus rhamnosus is more, and a sentence or two sentences to show their benefits is sufficient. So, author should add supplement the immune background or other content mentioned in this article based on your manuscript.

From Authors:

In response to the recommendation, we have removed the majority of the description for the strain-specific immune functions of L. rhamnosus and added other background information needed for understanding probiotic-mediated immune responses in the introduction.

  1. In the section of results, the author only needs to describe the changes in the results, without excessive discussion or introduction of knowledge background. Please simplify.

From Authors:

The authors have carefully considered the recommendation and have revised their manuscript to minimize unnecessary descriptions in the result section. The remaining information is intended to provide a prompt and helpful explanation for the readers.

  1. The discussion is a further analysis of the results, not a repetitive description of the results. Please correct.

From Authors:

The authors have revised their manuscript to provide a more analytical analysis in the discussion section, without giving redundant explanations that were already described in the results section.
